# Carbocyanine-Based Optical Sensor Array for the Discrimination of Proteins and Rennet Samples Using Hypochlorite Oxidation

**DOI:** 10.3390/s23094299

**Published:** 2023-04-26

**Authors:** Anna V. Shik, Irina A. Stepanova, Irina A. Doroshenko, Tatyana A. Podrugina, Mikhail K. Beklemishev

**Affiliations:** Department of Chemistry, M.V. Lomonosov Moscow State University, GSP-1, Leninskie Gory, 1–3, 119991 Moscow, Russia; shik.1966@mail.ru (A.V.S.); stepanovamsu@mail.ru (I.A.S.); doroshenkoiran@gmail.com (I.A.D.); podrugina@mail.ru (T.A.P.)

**Keywords:** sensor array, absorbance, fluorescence, fingerprinting, protein, redox reaction, carbocyanine dye, hypochlorite, rennet, linear discriminant analysis

## Abstract

Optical sensor arrays are widely used in obtaining fingerprints of samples, allowing for solutions of recognition and identification problems. An approach to extending the functionality of the sensor arrays is using a kinetic factor by conducting indicator reactions that proceed at measurable rates. In this study, we propose a method for the discrimination of proteins based on their oxidation by sodium hypochlorite with the formation of the products, which, in turn, feature oxidation properties. As reducing agents to visualize these products, carbocyanine dyes IR-783 and Cy5.5-COOH are added to the reaction mixture at pH 5.3, and different spectral characteristics are registered every several minutes (absorbance in the visible region and fluorescence under excitation by UV (254 and 365 nm) and red light). The intensities of the photographic images of the 96-well plate are processed by principal component analysis (PCA) and linear discriminant analysis (LDA). Six model proteins (bovine and human serum albumins, γ-globulin, lysozyme, pepsin, and proteinase K) and 10 rennet samples (mixtures of chymosin and pepsin from different manufacturers) are recognized by the proposed method. The method is rapid and simple and uses only commercially available reagents.

## 1. Introduction

Optical sensor arrays contain several cross-reactive sensors that respond differently to the samples of different compositions. Colorimetric and fluorometric sensor arrays have gained popularity due to their simplicity, easy operation, low cost, and versatile ability to solve tasks of quantitative and qualitative analysis. Optical array fingerprinting methods can assist in solving tasks of classification, reveal adulteration, identify manufacturer, detect impurities, and solve problems of medical diagnostics [1,2,3].

A popular task of optical fingerprinting is the recognition and discrimination of proteins, which can be based on several principles. The first approach is based on the competition of proteins to be detected with an enzyme for binding with nanostructures: gold nanoparticles (AuNPs) [4], DNA strands or aptamers adsorbed on AuNPs, which allowed for the recognition of 7–12 individual proteins [5,6], discrimination between cells [5], or the differentiation between albumin and globulin in human serum [6]. Competition of proteins with DNA was also used in the paper [7]. 

The different ability of proteins to cause aggregation of nanoparticles was used for their detection with colorimetric signal measurement. This strategy was realized by using AuNPs stabilized with surfactants [8,9] and Zr(IV) metal-organic framework (MOF) in combination with DNA-decorated AuNPs [10]. Fluorometry was also used, although less frequently; the quantum yield of fluorophore changed owing to interactions with proteins by multiple mechanisms [11,12,13]. The nanoparticles included nano-graphene oxide [11], fluorescent polymers [12], and carbon dots [13]. The named studies allowed for the recognition of a number of proteins in water and biological fluids. Optical array-based sensing of proteins has been considered in reviews [1,2,14,15].

A reaction-based approach to protein sensing has been proposed in papers [7,16,17]. The proteins moderated the catalytic activity of a Zr(IV)-based MOF [16], AuNPs [17], and Au, Ag, and Pd nanoparticles deposited on PPy@MoS_2_ nanostructures [7]. The methods are powerful: for example, the array [7] can identify proteins and their mixtures in human serum, identify heat-denatured proteins, determine individual proteins at nanomolar level, discriminate oral bacteria and clinical cancer samples. 

All reaction-based sensing studies [7,16,17] utilized oxidation of 3,3′,5,5′-tetramethylbenzidine (TMB) by hydrogen peroxide as an indicator process. We propose to broaden the set of reactions used for this purpose. In papers [18,19,20], we conducted oxidation of various carbocyanine dyes that resulted not only in a color change but also in fluorescence fading in visible and near IR regions. The components of the sample can alter the reaction rate by various mechanisms, including complexation of organic molecules with the metal ion catalyst. This strategy was evaluated in the recognition of sulfanylamides [18], discrimination of pharmaceuticals [19], and estimation of the doses absorbed by irradiated food [20]. Reaction-based sensing is an innovative technology with good prospects. 

In this study, we propose a different approach for the discrimination of proteins based on the treatment of the sample with hypochlorite. ClO^–^ ion is a strong oxidizing agent that acts primarily on the protein side-chains and the N-terminal α-amino groups, but partial cleavage of the protein backbone also happens [21,22]. 

Many protein oxidation products, in turn, also exhibit oxidizing properties. Some of them are believed to be chloramines (R–NHCl, R–NCl_2_) formed from the reaction of HOCl with the amine groups [21,22]. Due to different amounts of reactive groups, different proteins are expected to consume different amounts of hypochlorite and yield products of different oxidation capacity. The addition of a reactive reducing agent, such as a carbocyanine dye, can visualize the oxidation products, which are expected to be unique for each protein. A pre-oxidation technique based on chromic acid was used for the detection and identification of volatile organic compounds [23]. Treatment of samples with hypochlorite have not been used before for the discrimination purposes.

Many kinds of fermented milk and dairy products are manufactured using rennet, a group of gastric proteinases isolated from calf stomach; the two major rennet enzymes are pepsin and chymosin (rennin) [24]. Rennet formulations are marketed worldwide, but the enzyme activity and the properties of the product may vary depending on the brand, manufacturer, storage conditions, and other factors. To combat fraud (for example, sale of vegetable pepsin instead of animal one) and to be able to distinguish between rennets produced by different manufacturers, it is desirable to have a method for the discrimination of rennet samples. We have not found such methods in the literature. In this work, we demonstrate the possibility of recognizing six model proteins (bovine and human albumins, lysozyme, γ-globulin, pepsin, and proteinase K) and distinguishing 10 different brand rennet enzymes.

## 2. Materials and Methods

### 2.1. Reagents

The following model proteins were studied: bovine serum albumin (BSA), Sigma (Taufkirchen, Germany), CAS No 9048-46-8; albumin from human serum (HSA), Sigma, CAS No 70024-90-7; γ-globulin from human blood, Fluka (Buchs, Switzerland), CAS No 9007-83-4; pepsin from porcine gastric mucosa, Sigma, CAS No 9001-75-6; proteinase K from *Tritirachium album*, Sigma, CAS No 39450-01-6; lysozyme from chicken egg white, Sigma–Aldrich, CAS No 12650-88-3. The proteins were dissolved at 3 g/L in phosphate buffer (pH 7.35, 1/15 M). 

All rennet samples were acquired from Zdoroveevo (Moscow, Russia, https://www.zdoroveevo.ru, accessed on 20 March 2023). The samples (Table 1) were from different enzyme sources and had various chymosin/pepsin ratios. Sodium chloride was used as an excipient in all samples. The actual content of protein in the formulations estimated by us using the Bradford method (Section 2.4) is also shown in Table 1.

Carbocyanine dye Cy5.5 (CAS no. 1449612-07-0, Figure 1; its more common abbreviation is Cy5.5-COOH) was purchased from Lumiprobe (Hunt Valley, MD, USA, https://www.lumiprobe.com/, accessed on 20 March 2023) and dissolved in 95% ethanol (Bryntsalov-A, Moscow, Russia) to obtain a solution with a concentration of 1 g/L. On the day of experiments, a colloidal solution (0.017 g/L) was prepared by a 60-fold dilution of the stock solution with water. Carbocyanine dye IR-783, CAS no. 115970-66-6, is also commercially available, but for this study, it was synthesized by the authors according to a reported procedure [25]. The spectral data for IR-783 are provided in the Appendix A. The dye was dissolved in ethanol to 1 g/L and diluted with water 10-fold to obtain a working 0.1 g/L solution.

### 2.2. Instrumentation

Protein oxidation and subsequent indicator reactions were conducted in 96-well polystyrene plates (Nunc F96 MicroWell, white, Thermo Scientific, Waltham, MA, USA, cat. No. 136101, or Sovtech, Novosibirsk, Russia, cat. No. M-018). UV-vis spectra were recorded using an SF-102 spectrophotometer (Interfotofizika, Moscow, Russia) in quartz cells with an optical path length of 1 or 0.2 cm. Fluorescence spectra were obtained using a Fluorat-02 Panorama spectrofluorometer (Lumex, St. Petersburg, Russia). NIR fluorescence of the 96-well plates was monitored using a homemade NIR visualizer containing 11 red LEDs (660 nm) with a power of 3 W as a light source and a Nikon D80 camera with a light filter cutting off visible light up to 700 nm [26]. Visible-light photographs were captured with a Google Pixel 2 smartphone camera or Visualizer 2 (Camag, Muttenz, Switzerland). This instrument was also used to obtain fluorescence images with the excitation at 254 and 365 nm, the two built-in wavelengths of Camag Visualizer 2.

### 2.3. General Procedures

As is typical for such an experiment, a 10 µL portion of a protein sample or model protein solution was placed into a well of the plate. An amount of 30 µL of 5 mM NaOCl solution was pipetted, followed by 30 µL of 0.1 M acetate buffer, 170 µL of water (to obtain the final volume of the mixture equal to 300 µL), and finally a 60 µL portion of a dye solution (0.1 g/L in water for IR-783 or 0.017 g/L for Cy5.5). To fill multiple wells, 8-channel pipettes were used. Each sample or a model protein solution was measured in 6 replicates (6 wells of the plate were filled). The plate with the reaction mixture was photographed every several minutes to obtain four kind of images: (1) visible light absorbance (to be more precise, total light intensity, which is the result of absorption, scattering by colloid and reflection by the walls of the cell) by a smartphone camera or by the Camag visualizer; (2) and (3) fluorescence under the excitation by UV light (254 and 365 nm) in the Camag visualizer; and (4) near-infrared (NIR) fluorescence by red light (660 nm) excitation in the home-made visualizer. For quantitative measurements, two images at a time were obtained with the plate rotated 180° to compensate for its non-uniform illumination; the intensities of the two images were averaged.

### 2.4. Determination of Protein in Rennet Samples

To determine protein content in the rennet samples, Coomassie brilliant blue G-250 (CBB) method [27] was used with modifications. Two milligrams of CBB (Serva, Heidelberg, Germany) were dissolved in 1 mL of concentrated H_3_PO_4_, and 4 mL of ethanol was added to give a green-blue solution. To a well of a 96-well plate, 100 µL of the sample or a protein calibration solution and 30 µL of the 0.4 g/L CBB solution were added and the plate was photographed by the cell phone camera. The blank runs were colored green, and the presence of protein added blue color. The calibration was constructed for pepsin (Sigma) in the concentration range of *c* = 0.03–0.3 g/L. After digitization of the photographic images, the *I* = B − G − R function of intensities was calculated; it was found to produce a regression (which was more linear than a function of the individual R, G, B values) with an equation *I* = 133.0*c* + 7.1 and a correlation coefficient of 0.985 (for a total number of 8 points). The concentration of protein in samples was estimated using the obtained calibration.

### 2.5. Data Processing

The photographic images were digitized using the ImageJ 2.0.0-rc-61/1.51n software (Fiji) to determine mean intensities corresponding to individual wells. RGB splitting was performed to obtain red, green, and blue channel intensities (R, G, B) for the visible light reflection images; for fluorescence, only the overall intensity was measured. The intensities were presented as data tables with columns corresponding to images taken at different times; the visible and fluorescence images for both dyes were combined. The resulting number of columns in a table corresponding to one experiment was up to 49. The rows of the data table represented model protein solutions or rennet samples of a maximum number of 11 (together with the blank); accounting for the 6 replicates for each sample, the maximum number of rows was 66. Appendix A shows an example of a data table.

The data were subjected to principal component analysis (PCA) and linear discriminant analysis (LDA) using the XLSTAT (Addinsoft, New York, NY, USA) add-on for Microsoft Excel. PCA is an unsupervised technique in which the software is not informed by the user about the belonging of the samples to particular groups (classes). The results of PCA are visualized as score plots with principal components PC1 and PC2 (or PC1 and PC3) as coordinates. Each run (i.e., a replicate, or a well of a plate) is represented by a data point in the score plots. Confidence ellipses in the graphs were constructed using an 82% confidence interval.

The accuracy of discrimination of the model proteins or rennet samples was assessed by LDA using a cross-validation procedure in the XLSTAT software. The purpose of cross-validation is to test the ability of the XLSTAT model to predict new data. Validation divides data into two segments: one is used to train the model and the other is used to validate it. Cross-validation leaves several observations out of the training sample set and allows getting the prediction result for these observations using the current model; this operation is repeated for each observation. The averaged percentage correctness of the assignment of observations to samples was taken as a measure of discrimination accuracy. 

The following conditions for LDA analysis were found the most appropriate for this study: within-class covariance matrices were assumed to be equal; the stepwise (backward) model selection was activated with the threshold value of 0.05 to include the variable and 0.10 to leave out. A significance level of 5% was used in this study.

## 3. Results and Discussion

### 3.1. Development of the Test

#### 3.1.1. Hypochlorite–Protein–Dye Interactions

In this study, protein sensing is based on hypochlorite oxidation reactions. When a certain amount of NaOCl (for example, 0.3 µmol) is mixed with a small amount of a carbocyanine dye (a few nmols), a nearly instantaneous reaction takes place with complete discoloration of the dye. However, when NaOCl is first mixed with a protein (0.15 µmol with respect to the amino acid unit), oxidation processes occur. When the dye is added to the obtained protein–NaOCl mixture, its absorbance gradually decreases in the region over 550–600 nm and increases below 550–600 nm (Figure 1a,b). Some protein oxidation products form a colloid which causes background absorption over the whole spectrum, more intense at lower wavelengths. In the fluorescence spectra, the emission intensity rapidly decreases within the first few seconds, after which it slowly fades out (Figure 1c,d). This behavior indicates the formation of protein oxidation products that, in turn, exhibit oxidation properties with respect to the dye. These compounds, probably chloramines and others, can oxidize the dyes with color change and fluorescence fading (again: if there is no protein in the solution, the dye oxidizes nearly instantly).

The subsequent experiments were conducted in 96-well plates, which ensured high sample throughput. The color and fluorescence changes with time were followed photographically, and the intensities obtained from the photographic images were supposed to reflect the absorbance and fluorescence characteristics of the reaction mixtures. More precisely, the visible images involve the light that has passed through the solution and reflected from the plate walls; in addition, there may be some light scattering on the colloidal products which affects the overall signal intensity. However, our task was neither measuring absorbance nor determining the concentration of the dyes. Rather, we had to gain a different response from solutions of different compositions, which was successfully solved by using the overall signal: the total light intensity, which is the result of absorption, scattering by the colloid and reflection by the walls of the cell. The only concern should be repeatability, which is ensured by precise pipetting of solutions and observing the mixing order. Further below in the text, we only mention absorption for the sake of brevity. The situation is similar with fluorescent images. 

As shown in Figure 2 for one of the model proteins, the most noticeable changes take place during the first 5–10 min, although some changes in intensity are traceable up to 1 h. 

#### 3.1.2. Mixing order in NaOCl–Protein–Dye System

The reagent mixing order is important for obtaining kinetics favorable for using the reaction-based fingerprinting method. It was found that oxidation of the dye without protein or the addition of NaOCl as the last component resulted in nearly instantaneous discoloration of the dyes. Measurable reaction rates were achieved after contacting the hypochlorite with the protein for some time. For these conditions, two mixing orders were studied: (1) the oxidant was mixed with the protein and then the other components were added (see the details in the caption to Figure 3a), and (2) the protein was added to the well, diluted by adding water and buffer, and finally hypochlorite and the dye were added (Figure 3b). In the latter case, the oxidant did not completely react with the protein (since the solution was more diluted than in order 1), and instead, it oxidized the dye more rapidly. In Figure 3b, it can be seen that the signal has already changed by the first measurement (at about 0.5 min after the reaction started) and does not significantly change afterwards. In Figure 3a, the situation is more favorable: we can observe changes in the signal with time. We can also see that the oxidation of the dye without protein is nearly instantaneous for any mixing order (completed at 0.5 min with no further signal change). An almost complete absence of changes in absorbance over time for different pH values is also illustrated in Appendix A. These observations confirm the oxidation of the dyes with the protein oxidation products.

The time between mixing NaOCl + protein and adding the dye was studied for the 1st mixing order, and it was found that within 0.5–12 minb the resulting absorbance and fluorescence intensities did not change. This implies that the protein oxidation reactions occur very fast.

#### 3.1.3. Effect of pH on the Signal

A hypochlorite ion is more reactive than the acid (HOCl) and reacts most readily with protonated amine groups (RNH_3_^+^), which makes the oxidation rate maximal at intermediate pH values [21,22]. We have studied the dependence of the signal upon pH for the two dyes. At the pH values of ≤1 and >10, the dye oxidation was slow; oppositely, in the pH range of 3.0–9.2, more pronounced kinetic behavior was observed (Figure 4). For example, at pH 5.3, the discoloration rates of Cy5.5 for various model proteins are much higher (on average eight times) than at pH 1.0 (Table 2). We believe that more pronounced kinetics is a more favorable factor for discrimination of proteins, and pH 5.3 was selected for use in the entire work.

#### 3.1.4. Effect of Hypochlorite/Protein Ratio

The effect of the hypochlorite amount was studied by titration of a model protein with NaOCl (Figure 5). To ensure proper oxidation, protein was first mixed with NaOCl, and then the dye solution was added. With no protein in the system, an instantaneous reaction took place with complete discoloration of the dye and the fluorescence fading. This happened at ≥15 µL of 5 mM NaOCl per well (0.15 µmol), and slow oxidation of the dye was observed for 2–10 µL NaOCl. In the presence of protein, instant dye discoloration is only observed at 90 µL of NaOCl, while slower oxidation is observed for the other amounts of NaOCl at 1–4 min for visible images and much longer for the NIR images. Moreover, for long reaction times (16–43 min), it can be noticed that the dye discoloration proceeds for lower quantities of NaOCl (2–10 µL) that caused a faster reaction than in the blank runs. This fact can be only explained by higher reactivity of the protein oxidation products with respect to dye discoloration than original hypochlorite at the longer reaction times. These observations confirm the key role of the protein oxidation products that have oxidant properties.

As a result, we selected the following conditions for protein sensing: at first, protein and hypochlorite were mixed in about 2:1 molar ratio with respect to the amino acid unit, and pH 5.3 acetate buffer was added followed by the carbocyanine dye (in 0.05–0.30 molar ratio to hypochlorite, depending on the dye), after which all possible optical signals were measured as a function of time.

### 3.2. Discrimination of Model Proteins

Six proteins were selected as models to evaluate the discriminating power of the proposed system: BSA, HSA, γ-globulin, lysozyme, pepsin, and proteinase K. The composition and structure of proteins are supposed to determine their reactivity towards hypochlorite. The key idea of discriminating proteins using hypochlorite oxidation is that the nature and reactivity of their oxidation products (chloramines and others) should also strongly depend on the protein nature. It is assumed that different oxidation products will oxidize carbocyanine dyes at different rates, which has been confirmed in experiments (Figure 3a, Table 2). In Figure 6b,c, some additional data are given to show that model proteins demonstrate different reaction kinetic curves.

The difference in the properties of proteins is supposed to be determined by their amino acid composition and spatial structure. The groups most reactive with respect to hypochlorite oxidation include methylthio (methionine), mercapto (cysteine), heterocyclic nitrogen (tryptophan and histidine), phenolic hydroxyl (tyrosine), and free amino groups. The order of reactivity for the side-chains with respect to hypochlorite was found to be: Met > Cys >> Cystine ≈ His ≈ α-amino > Trp > Lys >> Tyr ≈ Arg > amide > Gln ≈ Asn [21]. The reaction with the most reactive amino acid, methionine, is nearly instantaneous: ≈10^8^ times higher than with H_2_O_2_ [22]. However, we observed no correlation between the amino acid composition (Appendix A) and the reactivity of proteins in the studied systems (Figure 6). Probably, structural features of the proteins play the leading role in their reactivity with respect to NaOCl and the reactivity of the oxidation products toward the dyes. 

Another possible reason of different kinetics and protein discrimination is protein–dye interaction. Particularly, free and protein-bound dyes may have different reactivities with respect to NaOCl and protein oxidation products, and dye-protein binding is dependent on the protein nature. It is difficult to directly confirm this suggestion because dye-protein complex and free protein will be oxidized by NaOCl in parallel. However, there are other indications of that feasibility; for example, in Figure 6b, we can see an increase in NIR emission for BSA and lysozyme (two upper curves) with respect to the blank run at the first minutes of the reaction, which can be explained by dye complexation with these proteins accompanied with an increase in the quantum yield. Thus, oxidation of dyes can be governed by different protein oxidation products, on the one hand, and on different dye–protein complexation, on the other hand.

The discrimination of model proteins was performed under the conditions selected in Section 3.1.4. The reaction mixtures (six replicate runs) were photographed in four spectral conditions up to 50 min from the reaction start, the images being captured every several minutes (an example is shown in Figure 6a). For IR-783 dye, not only the rate but also the nature of the oxidation products may depend on the nature of the initial protein, as we can judge from Figure 6a by the diversity of the color products. In total, 10 photographs were obtained with IR-783 and 9 ones with Cy5.5. These 19 images allowed us to obtain 49 data columns, because visible images were split into R, G, and B channels. These columns comprise the so-called full dataset (Appendix A). 

The PCA score plot was constructed in F1–F2 and F1–F3 principal components as coordinates, which showed good separation of the full dataset into groups corresponding to the individual proteins in the F1–F2–F3 three-dimensional space (Figure 7); proteinase and human serum albumin and globulin, not discriminated in the PC1–PC2 graph, are separated in the PC1–PC3 graph. 

The accuracy of discrimination of proteins was evaluated using LDA as a supervised technique. In performing the LDA procedure, the software sorts out the insignificant data columns (variables); for example, for the full dataset, only 24 out of 49 data columns have been found significant at a 5% threshold (Table 3, dataset No. 1). All proteins have been discriminated with 100% accuracy (measured as the number of correctly assigned samples to the total number of samples).

It is important to know whether any smaller dataset can provide an accurate discrimination, too. It would be desirable to omit some experiments (for example, the measurement of fluorescence under UV or red light excitation by taking the corresponding photographs). With that purpose, we manually excluded several data columns from the full dataset (Table 3, datasets No. 2–14). The dataset could be reduced to 20 columns (datasets No. 9 and 10), but the discrimination accuracy remained complete at 100%; the number of columns actually used by the software for the model creation was down to 14. It is important that some experiments could be completely omitted without compromising the discrimination accuracy: for example, UV-excited fluorescence study for IR-783 dye (datasets No. 2, 4, 7, 9, 10) and for Cy5.5 dye (datasets No. 3 and 4). Conversely, exclusion of NIR fluorescence data reduced accuracy from 100% to >90% (datasets No. 5, 6). Notably, a single Cy5.5 dye allowed us to discriminate the model proteins completely (dataset No. 7); the same was not true for IR-783 dye (dataset No. 6, accuracy 91%). However, using only one dye could be insufficient in solving more complicated problems such as discrimination of real-world samples. When the dataset was reduced even more significantly, for example, when only visible photographs were left for dyes IR-783 or Cy5.5 (datasets No. 11 and 13), the discrimination accuracy decreased to 91–94%, not allowing for the complete discrimination of the proteins (Table 3).

In Appendix A, the partial amino acid composition is given for the proteins used as models in this study. However, we have observed no connection between the amino acid composition and the pattern of oxidation of carbocyanine dyes (Figure 6a). It is probable that the amino acid percentage does not correlate with the ability of proteins to yield products serving as oxidizers after the reaction with NaOCl. It is possible that structural features have a greater effect on the reactivity of proteins with respect to NaOCl and the reactivity of the products toward the dyes

### 3.3. Detection of Various Protein Concentrations

The chemometric signal is supposed to be dependent on the concentration of proteins in solution. However, for low protein concentrations, too high of an amount of hypochlorite cannot be used in the protocol, since an excess of the oxidant will remain in the system and rapidly oxidize the dyes. For these experiments, as low as 5 µL of 5 mM NaOCl was used, and 0.3 to 30 µg of pepsin was introduced in the wells of the plate. The other conditions were kept as selected above. A total of 26 data columns were uploaded to the software, and 14 columns were used in the model development. The resulting score plot shows that the array easily detects 0.3 µg of the protein in the aqueous solution, and the other amounts differing by a factor of 3 (up to 30 µg) can also be estimated. The LDA results show that the discrimination accuracy between all these pepsin amounts is 100%. The linearity of detection was checked in coordinates: the pepsin amount in well-negative LDA factors F1 and F2 (Appendix A); for the second LDA factor (F2), the graph was linear for 0.3–3 µg of pepsin.

To determine the limit of detection, the range of low concentrations was examined (Appendix A). It can be seen that 0.03 µg/well of pepsin can be distinguished from the blank and from 0.3 µg/well, but not from 0.1 µg/well. Therefore, we can accept 0.03 µg as the LOD of pepsin for this method. In concentrations, this is 0.2 mg/L, or 6 nM (for the molar weight of pepsin of 34,500 Da), if a 150 µL portion of solution is used for analysis. The existing array-based methods allowed us to detect 50 nM [6], 9 nM [8], or 0.2 mg/L [13] of proteins in water. Consequently, the proposed method is comparable in sensitivity to the literature methods. 

### 3.4. Discrimination of Rennet Samples

The rennet samples purchased from one distributor were actually obtained from different physical sources and had different compositions (Table 1). It was a challenging task to discriminate between these samples, as they only contained two enzymes (chymosin and pepsin) from different biological species contained in different ratios. The rennet enzymes are marketed as mixtures with NaCl and contain from 6 to 15% of actual protein (Table 1). To perform the discrimination, we conducted two series of experiments: (1) with 1 g/L solutions of original formulations (not taking into account the actual protein content, which varied from 0.06 to 0.15 g/L) and (2) with more concentrated solutions, 3 g/L in total protein for each individual sample. 

The same protocol as described in Section 3.3 for the model proteins was applied to rennet samples. The visual appearance of the plates with the reaction mixtures are given in Appendix A. The discrimination results shown in Table 4 and Figure 8b show that complete discrimination is achieved for the full dataset and several shortened datasets. For example, it is possible to avoid using the UV excitation of fluorescence (254 and 365 nm) and capturing the corresponding images (dataset No. 4). Moreover, it is sufficient to use only visible light absorption images with the two dyes to obtain 100% accuracy (dataset No. 5). For all the datasets providing complete discrimination (Nos. 1–5), the LDA software selected no more than 18–21 data columns, which indicates that this amount of data is sufficient for the discrimination purposes. Further reduction of data impedes the discrimination quality (the accuracy falls below 100%) (datasets No. 6–9 in Table 4). For the less concentrated samples (1 g/L), the accuracy is lower (no complete discrimination is achieved) (Appendix A, Appendix A). However, if a practical problem is set, the samples are supposed to be available in quantities of at least milligrams, which would allow the preparation of more concentrated solutions.

Unfortunately, no correlation has been found between the signal and the reported pepsin/chymosin ratio in the samples (Table 1). Presumably, the source of the enzyme and impurities have a greater effect on the signal than the concentrations of pepsin and chymosin enzymes.

Presented results show the feasibility of rennet samples discrimination based on the sensitivity of the reaction-based array to proteins. Photographs of the plate in visible light taken during 40 min were sufficient to achieve 100% accuracy of discrimination; 21 data columns were shown to be minimally necessary by XLSTAT software. It was found that the use of a kinetic factor is an important advantage in the proposed array sensing method, as it is an additional parameter that is affected by sample composition.

## 4. Conclusions

To the best of our knowledge, this study is the first example of optical array sensing of proteins by using their oxidation. The proposed method proved to be efficient in recognition of model proteins as well as rennet enzyme samples. The difference in the properties of protein oxidation products is likely to play the key role in obtaining different responses from different proteins. Conditions for visualizing these differences by using carbocyanine dyes have been found. The use of a kinetic factor allows for the use of a small number of dyes in the array: remarkably, only one dye is sufficient to achieve complete discrimination of six model proteins or 10 rennet samples, and only the visible light images of the reaction mixtures with two dyes suffice to obtain 100% discrimination accuracy of 10 rennet samples.

The limitations of the proposed scheme are common with other pattern-based sensing methods: it is always necessary to process standards in parallel with the unknowns; when analyzing samples containing individual proteins, special effort is required to distinguish a change in the nature of a protein from a change in the concentration of the same protein; it is also problematic to automate the method or develop a version for field trials. The advantages of the proposed method are as follows: it is rapid and allows high sample throughput owing to the use of 96-well plates and the photographic principle of signal detection. The method is not relying on any sophisticated nanostructures and is believed to be reproducible; the protocol is simple and straightforward and uses commercially available reagents. Data are processed using standard chemometrics software. The prospects of the method include the possibility to recognize bacterial strains or to diagnose pathologies by fingerprinting of serum proteins.

## Data Availability

The data presented in this study are available as the electronic Appendix A.

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
