# Peer review of "Carbocyanine-Based Optical Sensor Array for the Discrimination of Proteins and Rennet Samples Using Hypochlorite Oxidation"

_sensors, 2023, doi:10.3390/s23094299_

Round 1

Reviewer 1 Report

Authors reported an interesting sensor array for the discrimination of proteins and rennet samples. The idea of indirect oxidation of dyes triggered by hypochlorite is nice. Details of methods are presented in a clear way. I recommend the major revision in addressing the following questions.

1. Chymosin and pepsin are major proteins in rennet sample, why did you exclude the chymosin in the model proteins?

2. The accuracy of discrimination should be further validated by test rennet samples rather than the training samples.

Author Response

Authors reported an interesting sensor array for the discrimination of proteins and rennet samples. The idea of indirect oxidation of dyes triggered by hypochlorite is nice. Details of methods are presented in a clear way. I recommend the major revision in addressing the following questions.

We are thankful to the Reviewer for the comments.

  1. Chymosin and pepsin are major proteins in rennet sample, why did you exclude the chymosin in the model proteins?

Reply. Pure chymosin (free from pepsin) was not readily available in the same large quantities as were used for other model proteins.

  1. The accuracy of discrimination should be further validated by test rennet samples rather than the training samples.

Reply. Discrimination accuracy can be tested by validation (using a test set and a training set) or by cross-validation (by using only training set from which a part is used as a test subset, then the operation is repeated for a different subset and so on). For simplicity, in the study we chose the latter method. To demonstrate that these methods are equal, we have compared the actual results shown by the cross-validation and validation methods using a separate test set. This was done for the rennet samples but not for the whole data set (because the results would be 100% accurate), but for IR-783 dye only (20 data columns). The size of the test set was taken as 16.7% of the whole data: one sample out of 6 was set aside and the model was trained on the rest 83.3% data. Then the results were predicted for the test set. The operation was repeated 6 times on a randomly selected test set. We obtained the accuracy by cross-validation of 89% and the accuracy by using the test set of (92±7)%, which is essentially the same. This example was to demonstrate that cross-validation can be used instead of validation using test sets. In many recognition tasks reported in literature cross-validation only is used.

Reviewer 2 Report

In this manuscript, Beklemishev et al proposed a method for the discrimination of proteins based on their oxidation by sodium hypochlorite with the formation of the products, which, in turn, feature oxidation properties. This work is meaningful and data solid, we recommend it publishable after addressing the following questions:

1.    As can be observed in Fig. 1 that sodium hypochlorite alone can oxidize the dye. How can we ensure accurate reaction between of lysozyme and sodium hypochlorite, or that the remaining sodium hypochlorite does not affect the spectrum of the dye+lysozyme+NaClO system?

2.    If there are multiple types of proteins in the sample, how can we achieve the detection of a single enzyme? 

3.    All experiments were conducted without error bars, which can lead to inaccurate experimental data.

Author Response

In this manuscript, Beklemishev et al proposed a method for the discrimination of proteins based on their oxidation by sodium hypochlorite with the formation of the products, which, in turn, feature oxidation properties. This work is meaningful and data solid, we recommend it publishable after addressing the following questions:

  1. As can be observed in Fig. 1 that sodium hypochlorite alone can oxidize the dye. How can we ensure accurate reaction between of lysozyme and sodium hypochlorite, or that the remaining sodium hypochlorite does not affect the spectrum of the dye+lysozyme+NaClO system?

Reply. The reaction occurs at first between the protein and NaOCl (not with the dye) since the dye is added only after the protein and NaOCl have reacted (and this reaction is almost instant, as shown under 3.1.2. Mixing order in NaOCl – protein – dye system). Now, to the second question, whether the remaining NaOCl affects the results: the amount of remaining NaOCl is only significant for the blank and low-protein samples, in these instances the indicatior reaction is rapid. In the presence of proteins NaOCl is consumed by them, and the subsequent oxidation of dyes proceeds with the chloramines (slower). For the samples that contain little amount of protein, the amount of NaOCl should be lowered (as we did in studying the dependence of the signal on the concentration of a protein under 3.2).

  1. If there are multiple types of proteins in the sample, how can we achieve the detection of a single enzyme? 

Reply. The detection of a single enzyme in mixture is not the task of this study. However, It is possible that some protein mixtures can be recognized. For instance, if there are proteins 1, 2, and 3 and their mixtures 1+2, 1+3, 2+3, it is highly likely that all the groups of signals can be separated in the scores plots. This is indirectly supported by the feasibility of recognition of 10 rennet enzyme mixtures in this study. If there are many proteins in the sample, the predominant ones can be possibly recognized.

  1. All experiments were conducted without error bars, which can lead to inaccurate experimental data.

Reply. The spectra (Fig. 1) are never given with error bars. The kinetic curves (Figs. 2, 3, 6c,d) have been shown only to give the idea of reaction kinetic behavior and not for quantitation purposes. The kinetic pH curves (Fig. 4) represent single experiments and for this reason do not contain the error bars. The remainder of the graphs are scores plots to which error bars are not applicable; the statistical treatment of data is provided by the built-in functions of the XLSTAT software. The experiments for discrimination purposes were all conducted in 6 parallel runs.

Reviewer 3 Report

In my opinion this manuscript is at the borderline between rejection and major, major revision. it requires more details regarding the instrumentation set up that need to be robust and artifact-free. Light scattering appears to affect significantly the measurements and their accuracy pending more towards rejection.

The NIR section would be better moved to SI as only the visual imaging is suggested in the conclusion.

I have attached several comments that need to be addressed thoroughly.

Reviewer 4 Report

Authors in conclusion mentioned that the work can find application in bacterial strain diagnosis or pathologies. Are there any studies performed in mixtures of proteins for instance combination of HSA and BSA, IgG or gamma-globulin?

What is the detection limit of the proposed method and linear range of the detection?

Also it is better to add discussion about the sensitivity of the reported methodology.

Author Response

  1. Authors in conclusion mentioned that the work can find application in bacterial strain diagnosis or pathologies. Are there any studies performed in mixtures of proteins for instance combination of HSA and BSA, IgG or gamma-globulin?

Reply. It is possible that simple protein mixtures can be recognized. For instance, if there are proteins 1, 2, and 3 and their mixtures 1+2, 1+3, 2+3 at comparable concentrations, it is highly likely that all the groups of signals can be separated in the scores plots. This is indirectly supported by the feasibility of recognition of 10 rennet enzyme mixtures in this study. If there are many proteins in the sample, the predominant ones can be possibly recognized. This is a subject for further studies.

  1. What is the detection limit of the proposed method and linear range of the detection?

Reply. We made a new experiment with the low quantities of pepsin to estimate the LOD. We have found that 0.03 µg/well of pepsin can be discriminated from the blank and from 0.3 µg/well, but not from 0.1 µg/well. Therefore, we can take 0.03 µg as the LOD with respect to pepsin for this method. In concentrations, this is 0.2 ppm, if a 150-µL portion of solution is used for analysis.

The linearity was checked in coordinates: pepsin amount in a well – negative LDA factors F1 and F2; for the second LDA factor (–F2) the graph was linear for 0.3 – 3 µg of pepsin. Larger quantities can be roughly estimated, up to 30 µg. This information has been added to text and SI.

  1. Also it is better to add discussion about the sensitivity of the reported methodology.

Reply. We compared the sensitivity of the proposed method to the existing methods reported in literature (an addition to section 3.3):

…we can accept 0.03 µg as the LOD of pepsin for this method. In concentrations, this is 0.2 mg/L, or 6 nM (for the molar weight of pepsin of 34,500 Da), if a 150-µL portion of solution is used for analysis. The existing array-based methods allowed to detect 50 nM [6], 9 nM [8], or 0.2 mg/L [13] proteins in water. Consequently, the proposed method is comparable in sensitivity to the literature methods.

Round 2

Reviewer 1 Report

I recommend the acceptance of publication in Sensors.

Author Response

We are thankful to the respected Reviewer for his work

Reviewer 3 Report

Provided the statements below are added to the revised manuscript to inform the reader of the aim and limitations of the presented test I am happy to recommend the final revised manuscript for publication in Sensor Journal.

“There is no doubt that light scattering can affect the signal intensity. But we should consider what do we mean by accuracy of measurements. If we wish to measure absorbance only, the light scattering will make it inaccurate. But our task was not measuring absorbance. In solving a practical task, we can measure total light intensity, which is the result of absorption, scattering by the colloid and reflection by the walls of the cell. The only concern should be repeatability, which is ensured by precise pipetting of solutions and observing the mixing order. We have added a statement to the text that in fact we measure not just absorbance but the overall intensity, which is determined by absorption, reflection and scattering.”

“Measuring the dye concentration was not our task, as well as determining the amount of the protein in colloidal form. Rather, we had to gain different response from solutions of different composition, and this task was successfully solved by using the overall signal. The correction for light scattering could be important in a theoretical study.”

Author Response

We combined the two paragraphs, having removed redundancy and polemical expressions, and included this text in section 3.1.1 as follows:

…there may be some light scattering on the colloidal products which affects the overall signal intensity. However, our task was neither measuring absorbance nor determining the concentration of the dyes. Rather, we had to gain different response from solutions of different composition, which was successfully solved by using the overall signal: the total light intensity, which is the result of absorption, scattering by the colloid and reflection by the walls of the cell. The only concern should be repeatability, which is ensured by precise pipetting of solutions and observing the mixing order.